# Distributed Multi-Player Bandits - a Game of Thrones Approach

**Ilai Bistritz**
Stanford University
bistritz@stanford.edu

**Amir Leshem**
Bar Ilan University
Amir.Leshem@biu.ac.il

## Abstract

We consider a multi-armed bandit game where $N$ players compete for $K$ arms for $T$ turns. Each player has different expected rewards for the arms, and the instantaneous rewards are independent and identically distributed. Performance is measured using the expected sum of regrets, compared to the optimal assignment of arms to players. We assume that each player only knows her actions and the reward she received each turn. Players cannot observe the actions of other players, and no communication between players is possible. We present a distributed algorithm and prove that it achieves an expected sum of regrets of near-$O\left(\log^2 T\right)$. This is the first algorithm to achieve a poly-logarithmic regret in this fully distributed scenario. All other works have assumed that either all players have the same vector of expected rewards or that communication between players is possible.

## 1 Introduction

In online learning problems, an agent needs to learn on the run how to behave optimally. The crux of these problems is the trade-off between exploration and exploitation. This trade-off is well captured by the multi-armed bandit problem, which has attracted enormous attention from the research community. Recently, there has been a growing interest in the case of the multi-player multi-armed bandit. In the multi-player scenario, the nature of the interaction between the players can take many forms. Players may want to solve the problem of finding the best mutual arm as a team [1–6], or may compete over the arms as resources they all individually require [7–19].

The idea of regret in the competitive multi-player multi-armed bandit problem is the expected sum of regrets and is defined as the performance loss compared to the optimal assignment of arms to players. The rationale for this notion of regret is formulated from the designer's perspective, who wants the distributed system of individuals to converge to a globally good solution.

Many works have considered a scenario where all the players have the same expectations for the rewards of all arms. Some of these works assume that communication between players is possible [10–12, 14, 19], whereas others consider a fully distributed scenario [7, 13, 15].

One of the main reasons for studying resource allocation bandits is their applications in cognitive radio or wireless networks in general. In these scenarios, the channels are interpreted as arms and the channel gains as the rewards. However, since users are scattered in space, the physical reality dictates that different arms have different expected channel gains for different players.

This essential generalization for a matrix of expectations introduces a famous combinatorial optimization problem known as the assignment problem [20]. Achieving a sublinear expected sum of regrets in a distributed manner requires a distributed solution to the assignment problem, which by itself has been explored extensively, e.g. [21, 22].

This generalization was first considered in [9], and later enhanced in [8], where an algorithm that achieves an expected sum of regrets of near-$O(\log T)$ was presented. However, this algorithm requires communication between players. It is based on the distributed auction algorithm in [21], which is not fully distributed. It requires that players can observe the bids of other players. This was possible in [8, 9] since it was assumed that the players could observe the actions of other players, which allows them to communicate by using the arm choices as a signaling method. In [19], the authors suggest an algorithm that only assumes that users can sense all the channels without knowing which channels was chosen by whom. This algorithm requires less communication than [8], but has no regret guarantees. In wireless networks, assuming that each user can hear all other transmissions (fully connected network) is very demanding in practice. In a fully distributed scenario, players only have access to their previous actions and rewards. However, to date there is no completely distributed algorithm that converges to the exact optimal solution of the assignment problem. The fully distributed multi-armed bandit problem remains unresolved.

Our work generalizes [7] for different expectations for different players and [8, 9, 19] for a fully distributed scenario with no communication between players.

Recently, very powerful payoff-based dynamics were introduced [23–25]. These dynamics only require each player to know her own action and the reward she received for that action. Specifically, the dynamics in [24] guarantee that the optimal sum of utilities strategy profile will be played a sufficiently large portion of the time, even if it is not a Nash equilibrium. The crucial issue of applying these results to our case is that they all assume interdependent games. In an interdependent game, each group of players can always influence at least one player from outside this group. In the multiplayer multi-armed bandit collision model, this does not hold. A player in a collision receives zero reward. Nothing that other players (who chose other arms) can do will change that.

In this paper, we suggest novel modified dynamics that behave similarly to [24], but in our non-interdependent game. Specifically, they guarantee that the optimal solution to the assignment problem is played a considerable amount of time. We present a fully distributed multi-player multi-armed bandit algorithm for the resource allocation and collision scenario, based on these modified dynamics. By fully distributed we mean that players only have access to their own actions and rewards. This is the first algorithm that achieves a poly-logarithmic expected sum of regrets, near-$O\left(\log^2 T\right)$, with a matrix of expected rewards and no communication at all between players.

## 2 Problem Formulation

We consider a stochastic game with the set of players $\mathcal{N} = \{1, ..., N\}$ and a finite time horizon $T$. The horizon $T$ is not known in advance by any of the players. The discrete turn index is denoted by $t$. The strategy space of each player is a set of $K$ arms with indices that are denoted by $i, j = 1, ..., K$. We assume that $K \geq N$. At each turn $t$, all players simultaneously pick one arm each. The arm that player $n$ chooses at time $t$ is $a_n(t)$ and the strategy profile at time $t$ is $\boldsymbol{a}(t)$. Players do not know which arms the other players chose, and need not even know how many other players are there.

Define the set of players that chose arm $i$ in strategy profile $\boldsymbol{a}$
$$\mathcal{N}_i(\boldsymbol{a}) = \{n \,|\, a_n = i\}. \tag{1}$$
Define the no-collision indicator of arm $i$ in strategy profile $\boldsymbol{a}$
$$\eta_i(\boldsymbol{a}) = \left\{ \begin{array}{ll} 0 & \left|\mathcal{N}_i(\boldsymbol{a})\right| > 1 \\ 1 & o.w. \end{array} \right. . \tag{2}$$
The instantaneous utility of player $n$ in strategy profile $\boldsymbol{a}(t)$ in time $t$ is
$$\upsilon_n(\boldsymbol{a}(t)) = r_{n,a_n(t)}(t)\, \eta_{a_n(t)}(\boldsymbol{a}(t)) \tag{3}$$
where $r_{n,a_n(t)}(t)$ is a random reward which is assumed to have a continuous distribution on $[0, 1]$. The sequence of rewards $\{r_{n,i}(t)\}_t$ of arm $i$ for player $n$ is i.i.d. ("in time") with expectation $\mu_{n,i}$.

Next we define the expected total regret, which we want our distributed algorithm to minimize.

**Definition 1.** Denote the expected utility of player $n$ in strategy profile $\boldsymbol{a}$ by $g_n(\boldsymbol{a}) = E\{\upsilon_n(\boldsymbol{a})\}$. The total regret is defined as the random variable
$$R = \sum_{t=1}^{T} \sum_{n=1}^{N} \upsilon_n(\boldsymbol{a}^*) - \sum_{t=1}^{T} \sum_{n=1}^{N} r_{n,a_n(t)}(t)\, \eta_{a_n(t)}(\boldsymbol{a}(t)) \tag{4}$$

where

$$\boldsymbol{a}^* = \arg\max_{\boldsymbol{a}} \sum_{n=1}^{N} g_n(\boldsymbol{a}).$$ (5)

The expected total regret $\bar{R} \triangleq E\{R\}$ is the average of (4) over the randomness of the rewards $\{r_{n,i}(t)\}_t$, that dictate the random actions $\{a_n(t)\}$.

The problem in (5) is no other than the famous assignment problem [20] on the $N \times K$ matrix of expectations $\{\mu_{n,i}\}$. In this sense, our problem is a generalization of the distributed assignment problem to an online learning framework.

Assuming continuously distributed rewards is well justified in wireless networks. Given no collision, the quality of an arm (channel) always has a continuous measure like the SNR or the channel gain. However, this assumption is only used in two arguments and can be easily replaced without changing the analysis in this paper. The first argument is that since the probability for zero reward in a non-collision is zero, players can safely rule out collisions in their estimation of the expected reward. In the case where the probability for a zero reward is not zero, we can assume instead that each player can observe her collision indicator in addition to her reward. Knowing whether other players chose the same arm is a very modest requirement compared to assuming that players can observe the actions of other players. The second argument is that the continuity of the rewards' distributions makes the solution of (5), with the estimated expectations, unique with probability 1. We can assume instead that $\{\mu_{n,i}\}$ are generated at random using a continuous distribution, so the optimal solution is unique with probability 1 (i.e., "for almost all games"), with arbitrary distributions for the rewards that have expectations $\{\mu_{n,i}\}$.

According to the seminal work in [26], the optimal regret of the single-player case is logarithmic; i.e., $O(\log T)$. Players do not help each other; hence, we expect the expected total regret lower bound to be logarithmic at best. The next proposition shows that this is indeed the case.

**Proposition 1.** *The expected total regret is at least* $\Omega(\log T)$.

*Proof.* See Section 8 (supplementary material). □

## 3 The Game of Thrones Algorithm

When all players have the same arm expectations, the exploration phase is used to identify the $N$ best arms. Once the best arms are identified, players need to coordinate to be sure that each of them will sit on a different "chair" (see the Musical Chairs algorithm in [7]). When players have different arm expectations, a non-cooperative **game** is induced where the estimated expected rewards serve as utilities. In this game, each player has a specific chair (**throne**) she must sit on to avoid causing a linear regret. This throne is the unique arm a player must play in the allocation that maximizes the sum of the expected rewards of all players. Any other solution will result in linear (in $T$) expected total regret. Note that our assignment problem has a unique optimal allocation with probability 1 (as shown in Lemma 4).

The total time needed for exploration increases with $T$ since the cost of being wrong becomes higher. When $T$ is known to the players, a long enough exploration can be accomplished at the beginning of the game. In order to maintain the right balance between exploration and exploitation when $T$ is not known in advance to the players, we divide the $T$ turns into epochs, one starting immediately after the other. Each epoch is further divided into three phases - exploration, Game of Thrones (GoT) and exploitation. During the exploration phase, players estimate the expected reward of each arm. The goal of the GoT phase is to let the players distributedly identify the optimal solution for the assignment problem on the estimated expected rewards from the exploration phase. It is done by playing a game with the estimated expectations as utilities, using random dynamics that probabilistically prefer strategy profiles with a higher sum of utilities. In the exploitation phase, each player plays the constant action she deduced from the GoT phase. The division into epochs is depicted in Fig. 1. The GoT Algorithm and GoT Dynamics are described in Algorithm 1 and Algorithm 2, respectively.

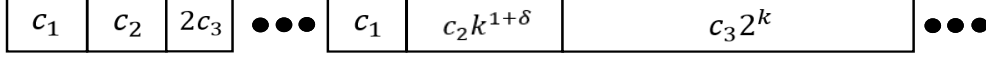

Figure 1: Epochs structure. Depicted are the first and the $k$-th epochs.

---

**Algorithm 1** Game of Thrones Algorithm

---

**Initialization** - Set $o_{n,i} = 0$ and $s_{n,i}(0) = 0$ for all $i$. Set $\delta > 0$, $0 < \rho < 1$ and $\varepsilon > 0$. Define $k_T$ as the index of the last epoch where the horizon is $T$.

**For each epoch** $k = 1, ..., k_T$

     1. **Exploration Phase** - for the next $c_1$ turns

         (a) Sample an arm $i$ uniformly at random from all $K$ arms.

         (b) Receive $r_{n,i}(t)$ and set $\eta_i(\boldsymbol{a}(t)) = 0$ if $r_{n,i}(t) = 0$ and $\eta_i(\boldsymbol{a}(t)) = 1$ otherwise.

         (c) If $\eta_i(\boldsymbol{a}(t)) = 1$ then update $o_{n,i} = o_{n,i} + 1$ and $s_{n,i}(t) = s_{n,i}(t-1) + r_{n,i}(t)$.

         (d) Estimate the expectation of the arm $i$ by $\mu_{n,i}^k = \frac{s_{n,i}(t)}{o_{n,i}}$, for each $i = 1, ..., K$.

     2. **GoT Phase** - for the next $c_2 k^{1+\delta}$ turns, play according to Algorithm 2 with $\varepsilon$ and $\rho$.

         (a) Starting from the $d_g = \lceil \rho c_2 k^{1+\delta} \rceil$-th turn inside the GoT Phase, keep track on the number of times each action was played that resulted in being content

$$F_t^n(i) = \sum_{l=d_g}^{c_2 k^{1+\delta}} I(a_n(l) = i, M_n(l) = C) \tag{6}$$

         where $I$ is the indicator function.

     3. **Exploitation Phase** - for the next $c_3 2^k$ turns, play

$$a_n^k = \arg\max_i F_t^n(i) \tag{7}$$

**End**

---

**Algorithm 2** Game of Thrones Dynamics

---

**Initialization** - Let $c \geq N$. Each player $n$ has a personal state $M_n$, either content $C$ or discontent $D$, which determines her mixed strategy. Each player also keeps a baseline action $\bar{a}_n$ and her utility is $u_n$. Denote $u_{n,\max} = \max_{\boldsymbol{a}} u_n(\boldsymbol{a})$.

**In each turn during the GoT Phase**

     • A content player chooses an action according to

$$p_n^{a_n} = \begin{cases} \frac{\varepsilon^c}{|\mathcal{A}_n| - 1} & a_n \neq \bar{a}_n \\ 1 - \varepsilon^c & a_n = \bar{a}_n \end{cases}. \tag{8}$$

     • A discontent player chooses an action uniformly at random; i.e.,

$$p_n^{a_n} = \frac{1}{|\mathcal{A}_n|}, \forall a_n \in \mathcal{A}_n. \tag{9}$$

The transitions between $C$ and $D$ are determined as follows:

     • If $\bar{a}_n = a_n$ and $u_n > 0$, then a content player remains content with probability 1

$$[\bar{a}_n, C] \to [\bar{a}_n, C] \tag{10}$$

     • If $\bar{a}_n \neq a_n$ or $u_n = 0$ or $M_n = D$, then ($C/D$ denoting either $C$ or $D$)

$$[\bar{a}_n, C/D] \to \begin{cases} [a_n, C] & \frac{u_n}{u_{n,\max}} \varepsilon^{u_{n,\max} - u_n} \\ [a_n, D] & 1 - \frac{u_n}{u_{n,\max}} \varepsilon^{u_{n,\max} - u_n} \end{cases}. \tag{11}$$

**End**

---

In this paper, we prove the following main result.

**Theorem 1** (Main Theorem). *Assume that the rewards $\{r_{n,i}(t)\}_t$ are independent in $n$ and i.i.d. in time $t$, with continuous distributions on $[0,1]$ with positive expectations $\{\mu_{n,i}\}$. Let the game have a finite horizon $T$, unknown to the players. Denote the optimal objective by $J_1 = \max_{\boldsymbol{a}} \sum_{n=1}^{N} g_n(\boldsymbol{a})$ and the second best one by $J_2$. Let each player play according to Algorithm 1, with a small enough $\varepsilon$, exploration phase length of $c_1 > \frac{16N^2K}{(J_1-J_2)^2}$ and $\delta > 0$. Then, for large enough $T$, the expected total regret is upper bounded by*

$$\bar{R} \le 3c_2 N \log_2^{2+\delta}\left(\frac{T}{c_3}+2\right) = O\left(\log^{2+\delta} T\right). \tag{12}$$

*Proof.* Let $\delta > 0$. Denote the number of epochs that start within $T$ turns by $E$. Since

$$T \ge \sum_{k=1}^{E-1}\left(c_1 + c_2 k^{1+\delta} + c_3 2^k\right) \ge c_3\left(2^E - 2\right) \tag{13}$$

$E$ is upper bounded by $E \le \log_2\left(\frac{T}{c_3}+2\right)$. Denote by $P_{e,k}$ and $P_{c,k}$ the error probabilities of the exploration and GoT phases of epoch $k$ respectively. Observe that if none of these errors occurred, the optimal solution to (5) is played in the $k$-th exploitation phase, which adds no additional regret to the total regret. We will prove in Lemma 2 and Lemma 5 that $P_{e,k} \le 4K^2 e^{-k}$ and $P_{c,k} \le A_0 e^{-\frac{c_2(1-\rho)}{1728 T_m\left(\frac{1}{8}\right)}k^{1+\delta}}$, where $A_0$ is a constant and $T_m\left(\frac{1}{8}\right)$ is the mixing time of the Markov chain of the GoT Dynamics. Note that $T_m\left(\frac{1}{8}\right)$ depends on $N, K$ and $\varepsilon$, so there exists a $k_0$ such that for all $k > k_0$ we have

$$e^{-\frac{c_2(1-\rho)}{1728 T_m\left(\frac{1}{8}\right)}k^{\delta}} < \frac{1}{2}. \tag{14}$$

We now bound the expected total regret of epoch $k > k_0$, denoted by $\bar{R}_k$, as follows

$$\bar{R}_k \le \left(c_1 + c_2 k^{1+\delta}\right)N + \left(4K^2 e^{-k} + A_0 e^{-\frac{c_2(1-\rho)}{1728 T_m\left(\frac{1}{8}\right)}k^{1+\delta}}\right)c_3 2^k N \le$$

$$c_1 N + 2A_0 c_3 N \beta^k + c_2 k^{1+\delta} N \tag{15}$$

for some constant $\beta < 1$. We conclude that, for some additive constant $C$,

$$\bar{R} = \sum_{k=1}^{E} \bar{R}_k \underset{(a)}{\le} C + 2c_2 N \sum_{k=k_0+1}^{E} k^{1+\delta} \le C + 2c_2 N E^{2+\delta} \underset{(b)}{\le} C + 2c_2 N \log_2^{2+\delta}\left(\frac{T}{c_3}+2\right) \tag{16}$$

where (a) follows since completing the last epoch to a full epoch increases $\bar{R}_k$, and (b) is (13).

$\square$

If either the exploration or the GoT phases fail, the regret becomes linear with $T$. Like many other online learning algorithms, we avoid a linear expected regret by ensuring that the error probabilities vanish with $T$. By using instead a single epoch with a constant duration for the first two phases, we obtain that with high probability (in $T$) our algorithm achieves a constant regret (as in [7]). However, our main result is formulated using the more conservative formulation of the expected regret.

## 4 Exploration Phase - Estimation of the Expected Rewards

In this section, we describe the exploration phase, and analyze its addition to the expected total regret. At the beginning of the game, players still do not have any evaluation of the $K$ different arms. They estimate these values on the run, based on the rewards they get. We propose a pure exploration phase where each player picks an arm uniformly at random, similar to the one suggested in [7]. Note that in contrast to [7], we do not assume that $T$ is known to the players. Hence, the exploration phase is repeated in each epoch. In each epoch, only a constant number $c_1$ of turns is

dedicated to exploration. However, the estimation uses all the previous exploration phases, so that the number of samples for estimation grows linearly with time.

The estimation of the expected rewards is never perfect. Hence, the optimal solution to the assignment problem given the estimated expectations might be different from the optimal solution with the correct expectations. However, if the uncertainty of the true value of each expectation is small enough, we expect both of these optimal assignments to coincide. This is exactly the precision we require from the estimation, as formulated in the following lemma.

**Lemma 1.** *Assume that $\{\mu_{n,i}\}$ are known up to an uncertainty of $\Delta$, i.e., $|\hat{\mu}_{n,i} - \mu_{n,i}| \leq \Delta$ for each $n$ and $i$ for some $\{\hat{\mu}_{n,i}\}$. Denote the optimal assignment by $\boldsymbol{a_1} = \arg\max_{\boldsymbol{a}} \sum_{n=1}^{N} g_n(\boldsymbol{a})$ and its objective by $J_1 = \sum_{n=1}^{N} g_n(\boldsymbol{a_1})$. Denote the second best objective and the corresponding assignment by $J_2$ and $\boldsymbol{a_2}$, respectively. If $\Delta < \frac{J_1 - J_2}{2N}$ then*

$$\arg\max_{\boldsymbol{a}} \sum_{n=1}^{N} g_n(\boldsymbol{a}) = \arg\max_{\boldsymbol{a}} \sum_{n=1}^{N} \hat{\mu}_{n,a_n} \eta_{a_n}(\boldsymbol{a}) \tag{17}$$

*so that the optimal assignment does not change due to the uncertainty.*

*Proof.* See Section 8 (supplementary material). □

If the exploration phase is long enough, players know their arm expectations accurately enough with a very small failure probability. The following lemma concludes this section by providing an upper bound for the probability that the estimation for epoch $k$ failed.

**Lemma 2** (Exploration Error Probability). *Let $\{\mu_{n,i}^k\}$ be the estimated reward expectations using all the exploration phases up to epoch $k$. Denote $\boldsymbol{a}^* = \arg\max_{\boldsymbol{a}} \sum_{n=1}^{N} g_n(\boldsymbol{a})$ and $\boldsymbol{a}^{k*} = \arg\max_{\boldsymbol{a}} \sum_{n=1}^{N} \mu_{n,a_n}^k \eta_{a_n}(\boldsymbol{a})$. Also denote $J_1 = \sum_{n=1}^{N} g_n(\boldsymbol{a}^*)$ and the second best[1] objective by $J_2$. If the length of the exploration phase satisfies $c_1 > \frac{16N^2 K}{(J_1 - J_2)^2}$, then after the $k$-th epoch we have*

$$P_{e,k} \triangleq \Pr\left(\boldsymbol{a}^* \neq \boldsymbol{a}^{k*}\right) \leq 4K^2 e^{-k}. \tag{18}$$

*Proof.* See Section 8 (supplementary material). □

## 5 Game of Thrones Dynamics Phase

In this section we analyze the game of thrones (GoT) dynamics between players. These dynamics guarantee that the optimal state will be played a significant amount of time, and only require the players to know their own action and the received payoff on each turn. Note that these dynamics assume deterministic utilities. We use the estimated expected reward of each arm as the utility for this step, and zero if a collision occurred. This means that players ignore the numerical reward they receive by choosing the arm, as long as it is positive.

**Definition 2.** The game of thrones $G$ of epoch $k$ has the $N$ players of the original multi-armed bandit game. Each player can choose from among the $K$ arms, so $\mathcal{A}_n = \{1, ..., K\}$ for each $n$. The utility of player $n$ in the strategy profile $\boldsymbol{a} = (a_1, ..., a_N)$ is

$$u_n(\boldsymbol{a}) = \mu_{n,a_n}^k \eta_{a_n}(\boldsymbol{a}) \tag{19}$$

where $\mu_{n,a_n}^k$ is the estimation of the expected reward of arm $a_n$, from all the exploration phases that have ended, up to epoch $k$. Also denote $u_{n,\max} = \max_{\boldsymbol{a}} u_n(\boldsymbol{a})$.

Our dynamics belong to the family introduced in [23–25]. These dynamics guarantee that the optimal sum of utilities strategy profiles will be played a sufficiently large portion of the turns. However, they all rely on the following structural property of the game, called interdependence.

**Definition 3.** A game $G$ with finite action spaces $\mathcal{A}_1, ..., \mathcal{A}_N$ is interdependent if for every strategy profile $\boldsymbol{a} \in \mathcal{A}_1 \times ... \times \mathcal{A}_N$ and every set of players $J \subset N$, there exists a player $n \notin J$ and a choice of actions $\boldsymbol{a}'_J \in \prod_{m \in J} \mathcal{A}_m$ such that $u_n(\boldsymbol{a}'_J, \boldsymbol{a}_{-J}) \neq u_n(\boldsymbol{a}_J, \boldsymbol{a}_{-J})$.

Our GoT is not interdependent. To see this, pick any strategy profile $\boldsymbol{a}$ such that some players are in a collision while others are not. Choose $J$ as the set of all players that are not in a collision. All players outside this set are in a collision, and there does not exist any colliding player that the actions of the non-colliding players can make her utility non-zero.

The GoT Dynamics in Algorithm 2 modify [24] such that interdependency is no longer needed. Note that in comparison with [24], our dynamics assign zero probability that a player with $u_n = 0$ (in a collision) will be content. Additionally, we do not need to keep the benchmark utility as part of the state. A player knows with probability 1 whether there was a collision, and if there was not, she gets the same utility for the same arm. Our dynamics require that each player uses $c \geq N$. The number of players $N$ might be unknown. In this case, players can use $c \geq K$, since the number of arms is known and $K \geq N$ by definition of the problem.

The GoT dynamics induce a Markov chain over the state space $Z = \prod_{n=1}^{N} (\mathcal{A}_n \times \mathcal{M})$, where $\mathcal{M} = \{C, D\}$. The transition matrix of this Markov chain is denoted by $P^\varepsilon$. The following lemma characterizes the recurrence classes of the unperturbed chain $P^0$ (with $\varepsilon = 0$). In [24], interdependency was used to prove the same result. This is the sole reason interdependency was required in the first place. We provide an alternative proof that does not require interdependency but instead uses the fact that in our modified dynamics, no player can be content with $u_n = 0$. Note that this proof exploits the structure of the GoT, and cannot be applied to a more general game.

**Lemma 3.** *Denote by $D_0$ the set of all the discontent states (all players are discontent) and by $C_0$ the set of all singleton content states (all players are content). The recurrence classes of the unperturbed process $P^0$ are $D_0$ and all $z \in C_0$.*

*Proof.* In $P^0$, there is no path between the discontent states and the content ones. Moreover, all the discontent states are connected and all the content states are absorbing (i.e., singletons). Now assume there is a different recurrence class. In any state in this class, denoted $z_{C/D}$, not all the players are content, otherwise this is a $z \in C_0$ singleton. Denote one of the discontent players by $n$. Since she chooses her action at random, there is a positive probability that she will pick the same arm as any of the content players. By doing so, she changes the state of this player to discontent with probability 1. With $\varepsilon = 0$, every discontent player remains so with probability one. On the next turn, a discontent player may again choose the arm of a content player with a positive probability. By repeating this process, we conclude that there is a positive probability that all players become discontent. Hence, $z_{C/D}$ is connected to $D$ in $P^0$. We conclude that this different recurrence class is in fact connected to $D$, which is a contradiction. $\qquad\square$

The process $Z$ of the GoT dynamics is a regular perturbed Markov chain, defined as follows.

**Definition 4.** $P^\varepsilon$ is called a regular perturbed Markov Process if $P^\varepsilon$ is ergodic for all sufficiently small $\varepsilon > 0$ and for every $z, z' \in Z$ we have

$$\lim_{\varepsilon \to 0^+} P^\varepsilon_{zz'} = P^0_{zz'} \tag{20}$$

and if $P^\varepsilon_{zz'} > 0$ for some $\varepsilon > 0$ then

$$0 < \lim_{\varepsilon \to 0^+} \frac{P^\varepsilon_{zz'}}{\varepsilon^{r(z \to z')}} < \infty \tag{21}$$

for some real non-negative $r(z \to z')$ that is called the resistance of the transition $z \to z'$.

Next we define stochastic stability, which is a powerful convergence analysis tool.

**Definition 5.** Let $P^\varepsilon$ be regular perturbed Markov process and $\mu^\varepsilon$ its unique stationary distribution that exists for $\varepsilon > 0$. A state $z \in Z$ is stochastically stable if and only if

$$\lim_{\varepsilon \to 0^+} \mu^\varepsilon(z) > 0. \tag{22}$$

In [24], it is shown for their dynamics that only the states with the maximal sum of utilities are stochastically stable. For a small enough $\varepsilon$ the dynamics will visit the stochastically stable states

very often. However, there might be several stochastically stable states and the dynamics might fluctuate between them. Fortunately, in our case, as shown in the following lemma, there is a unique optimal state with probability one. For a small enough $\varepsilon$ the unique optimal state is played more than half of the times, which allows for the players to distributedly agree on the optimal solution. This uniqueness is due to the continuous distribution of the rewards that makes the distribution of the empirical estimation for the expectations continuous as well.

**Lemma 4.** *The optimal solution to $\max_{\boldsymbol{a}} \sum_{n=1}^{N} u_n(\boldsymbol{a})$ is unique with probability 1.*

*Proof.* First note that an optimal solution must not have any collisions, otherwise it can be improved since $K \geq N$. Let $\{\mu_{n,i}^k\}$ be the estimated reward expectations in epoch $k$. For two different solutions $\tilde{\boldsymbol{a}} \neq \boldsymbol{a}^*$ to be optimal, we must have $\sum_{n=1}^{N} \mu_{n,\tilde{a}_n}^k = \sum_{n=1}^{N} \mu_{n,a_n^*}^k$. However, $\tilde{\boldsymbol{a}}$ and $\boldsymbol{a}^*$ must differ in at least one assignment. Since the distributions of the rewards $r_{n,a_n}$ are continuous, so are the distributions of $\sum_{n=1}^{N} \mu_{n,a_n}^k$ (as a sum of the average of the rewards). Hence $\Pr\left(\sum_{n=1}^{N} \mu_{n,\tilde{a}_n}^k = \sum_{n=1}^{N} \mu_{n,a_n^*}^k\right) = 0$, and the result follows. $\qquad\square$

Next we show that only the unique optimal state is stochastically stable. This means that after enough time, the action that a player played most of the time is highly likely to be part of the unique optimal solution. This is crucial for the success of the exploitation phase.

**Theorem 2.** *Define $\boldsymbol{a}^{k*} = \arg\max_{\boldsymbol{a}} \sum_{n=1}^{N} u_n(\boldsymbol{a})$. Under the GoT dynamics, the unique stochastically stable state is $z^* = [\boldsymbol{a}^{k*}, C^N]$ with probability 1.*

*Proof.* See Section 8 (supplementary material). $\qquad\square$

Now we can prove the main lemma of this section that gives an upper bound for the probability that the GoT phase does not lead to the optimal solution.

**Lemma 5** (GoT Error Probability)**.** *Let $\delta > 0$. Define $\boldsymbol{a}^{k*} = \arg\max_{\boldsymbol{a}} \sum_{n=1}^{N} u_n(\boldsymbol{a})$ and $\widetilde{\boldsymbol{a}} = (\widetilde{a}_1, ..., \widetilde{a}_N)$ where $\widetilde{a}_n = \arg\max_{i} F_t^n(i)$ for all $n$. For a small enough $\varepsilon$, the error probability of the $k$-th GoT phase, which is the probability that another strategy profile than $\boldsymbol{a}^{k*}$ will be played in the exploitation phase, is bounded as follows*

$$P_{c,k} \triangleq \Pr\left(\widetilde{\boldsymbol{a}} \neq \boldsymbol{a}^{k*}\right) \leq A_0 e^{-\frac{c_2(1-\rho)}{1728 T_m\left(\frac{1}{8}\right)} k^{1+\delta}}. \tag{23}$$

*where $A_0$ is a constant with respect to $t$ (or $k$), and may depend on $N, K, \varepsilon$ and the initial state.*

*Proof.* See Section 8 (supplementary material). $\qquad\square$

## 6 Numerical Simulations and Practical Considerations

The total regret compares the sum of utilities to the ideal one that could have been achieved in a centralized scenario. With no communication between players and with a matrix of expected rewards, the gap to this ideal naturally increases. In this scenario, converging to the exact optimal solution might take a long time, even for the (unknown) optimal algorithm. Our main result provides theoretical guarantees for the asymptotic performance of our algorithm, which suggest that performance improves with time on its way to converge to the optimal solution. The simulations in this section complete the picture by showing how the sum of utilities behaves in the non-asymptotic regime.

We simulated a multi-armed bandit game with $\{\mu_{n,i}\}$ that are chosen independently and uniformly at random in $[0.05, 0.95]$. The rewards are generated as $r_{n,i}(t) = \mu_{n,i} + z_{n,i}(t)$ where $\{z_{n,i}(t)\}$ are independent and uniformly distributed on $[-0.05, 0.05]$ for each $n, i$.

In the simulations presented here we use $\delta = 0$ since it yields good results in practice. We conjecture that the bound (23) is not tight for our particular Markov chain and indicator function, since it applies for all Markov chains with the same mixing time and all functions on the states. This explains why modest choices of $c_2$ are large enough and the $k^\delta$ factor in the exponent is not needed in practice.

The lengths of the phases should be chosen so that the exploitation phase occupies most of the turns already in early epochs, while allowing for a considerable GoT phase. Note that the exploration $(c_1)$ is much easier than the GoT phase $(c_2)$ and achieves a good accuracy relatively fast. Hence we choose $c_1 = 1000, c_2 = c_3 = 6000$. We use $\rho = \frac{1}{2}$ in the simulations we present, since the performance is very similar for $\rho$ values not too close to zero or one. We use $c = N$, that gives the highest possible escape probability of $\varepsilon^c$ from a content state.

In Fig. 2, we present the sample mean of the accumulated sum of utilities $\sum_{n=1}^{N} \frac{1}{t} \sum_{\tau=1}^{t} u_n(\boldsymbol{a}(\tau))$ as a function of time $t$, averaged over 100 experiments. The performance was normalized by the optimal solution to the assignment problem (for each experiment). On the left graph we compare our sum of utilities for $N = K = 5$ to that of the selfish algorithm, reported to achieve good performance for this problem in [17], and to a random choice of arms. The selfish algorithm consists of each player playing a standard upper confidence bound (UCB) algorithm, treating collisions as any other value for the reward. Both algorithms perform much better than the random selection. Our sum of utilities is slightly better and is increasing with time. More importantly, our algorithm has provably performance guarantees while [17] have none. On its way to converge to the optimal solution, our algorithm performs very well straight from the beginning. While visiting near-optimal solutions inflicts linear regret at the beginning, it is very satisfying in practice considering that players cannot communicate and have a matrix of expected rewards. Similar results were obtained for different choices of $c_1, c_2, c_3$. On the right graph, we present the median and the best 90% of the sample mean of the sum of utilities for $K = N = 6$ and $\varepsilon = 0.01, 0.001, 0.0001$. It is evident that our algorithm behaves very similarly in all the 100 experiments, indicating that it is robust and rarely fails. Additionally, our algorithm behaves very similarly for a wide range of $\varepsilon$ values (two orders of magnitude). This supports the intuition that there is no threshold phenomenon on $\varepsilon$ (becoming "small enough"), since the dynamics prefer states with a higher sum of utilities for all $\varepsilon < 1$.

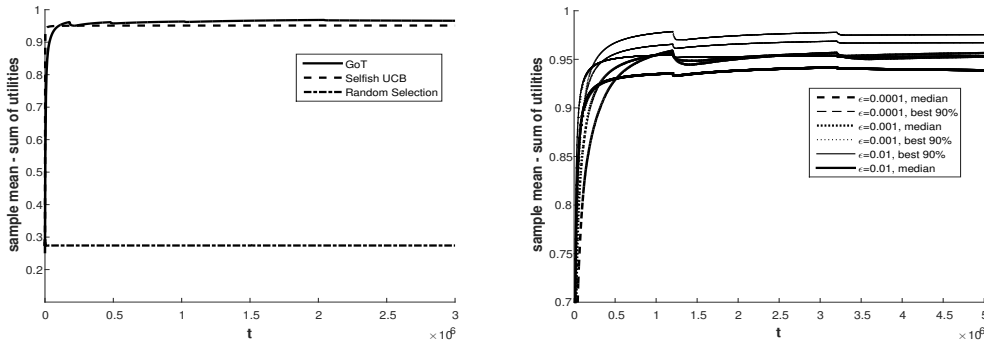

Figure 2: Sample mean of the sum of utilities as a function of time, averaged over 100 experiments.

## 7    Conclusions and Future Work

In this paper, we considered a multi-player multi-armed bandit game where players compete over the arms as resources. In contrast to all existing multi-player bandit problems, we allow for different arm expected rewards between players **and** assume each player only knows her own actions and rewards. We proposed a novel fully distributed algorithm that achieves a poly-logarithmic expected total regret of near-$O\left(\log^2 T\right)$ when the horizon $T$ is unknown to the players.

Our simulations suggest that tuning the parameters for our algorithm is a relatively easy task in practice. The algorithm designer can do so by simulating a random model for the unknown environment and varying the parameters, knowing that only a very slack accuracy is needed for the tuning.

It is still an open question whether the lower bound $\Omega\left(\log T\right)$ on the expected total regret is tight for a fully distributed algorithm.

Our game is not a general one but has a structure that allowed us to modify the dynamics such that the interdependence assumption can be dropped. We conjecture that the same structure can be exploited to accelerate the convergence rate of the GoT dynamics, specifically by relaxing the $c \geq N$ condition.

## Footnotes

[1]Note that this is the second best objective and not the second best allocation, so $J_2 < J_1$. If all allocations have the same objective then this Lemma trivially holds with $c_1 \geq 1$.

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
