[Supplementary Material]

# 8 Proofs

## 8.1 Proof Of Proposition 1

*Proof.* For $N = 1$, the result directly follows from [26]. Now we formally prove that more players cannot help. Assume that for $N > 1$ there is a policy that results in a better total expected regret than $\Omega(\log T)$. Hence, there must exist a player for which the personal regret is also better than $\Omega(\log T)$. This player, denoted player $n$, can simulate $N - 1$ other players and generate at random their expectations and rewards, all of which are independent of the actual rewards she receives. This player also simulates the policies for other players, and even knows when a collision occurred for herself and can assign zero reward in that case. Hence, simulating $N - 1$ fictitious players is a valid single player multi-armed bandit policy that violates the $\Omega(\log T)$ bound, which is a contradiction. We conclude that this bound is also valid for $N > 1$. $\qquad\square$

## 8.2 Proof Of Lemma 1

*Proof.* First note that an optimal solution must not have any collisions, otherwise it can be improved since $K \geq N$. Hence $J_1 = \sum_{n=1}^{N} \mu_{n,a_{1,n}}$. For all $n$ and $i$ we have $\hat{\mu}_{n,i} = \mu_{n,i} + z_{n,i}$ such that $|z_{n,i}| \leq \Delta$. In the perturbed assignment problem, $\boldsymbol{a}_1$ performs at least as well as

$$\sum_{n=1}^{N} \hat{\mu}_{n,a_{1,n}} = \sum_{n=1}^{N} \left( \mu_{n,a_{1,n}} + z_{n,i} \right) \geq \sum_{n=1}^{N} \mu_{n,a_{1,n}} - \Delta N \tag{24}$$

and any assignment $\boldsymbol{a} \neq \boldsymbol{a}_1$ performs at most as well as

$$\sum_{n=1}^{N} \hat{\mu}_{n,a_n} \eta_{a_n}(\boldsymbol{a}) = \sum_{n=1}^{N} \left( \mu_{n,a_n} + z_{n,i} \right) \eta_{a_n}(\boldsymbol{a}) \leq \sum_{n=1}^{N} \mu_{n,a_{2,n}} \eta_{a_{2,n}}(\boldsymbol{a}_2) + \Delta N. \tag{25}$$

Hence it follows that if $\Delta < \frac{J_1 - J_2}{2N}$ then for every $\boldsymbol{a} \neq \boldsymbol{a}_1$

$$\sum_{n=1}^{N} \hat{\mu}_{n,a_{1,n}} > \sum_{n=1}^{N} \hat{\mu}_{n,a_n} \eta_{a_n}(\boldsymbol{a}). \tag{26}$$

$\qquad\square$

## 8.3 Proof Of Lemma 2

*Proof.* According to [7, Lemma 1], for each $\Delta > 0$ and error probability $0 < P_{e,k} < 1$, after

$$T_0 = \frac{4K}{\Delta^2} \ln\left( \frac{4K^2}{P_{e,k}} \right) \tag{27}$$

turns of pure exploration, for all $n$ and all $i$ we have with a probability of at least $1 - P_{e,k}$ that

$$\left| \mu_{n,i}^k - \mu_{n,i} \right| \leq \Delta. \tag{28}$$

In [7], the formulation of the lemma is slightly different and states the probability of an $2\Delta$-correct ranking. However, the proof follows by showing (28). Note that we used the fact that

$$\Delta < \frac{J_1 - J_2}{2N} \leq \frac{N - 0}{2N} = \frac{1}{2} < 1$$

so

$$T_0 = \max\left\{ \frac{K}{2} \ln\left( \frac{2K^2}{P_{e,k}} \right), \frac{4K}{\Delta^2} \ln\left( \frac{4K^2}{P_{e,k}} \right) \right\} = \frac{4K}{\Delta^2} \ln\left( \frac{4K^2}{P_{e,k}} \right). \tag{29}$$

We conclude that after $T_0$ exploration turns, the error probability is at most $P_{e,k}$. Hence, if the exploration phase has a duration of at least $\frac{4K}{\Delta^2}$ turns we obtain

$$\frac{4K}{\Delta^2} k \leq c_1 k = \frac{4K}{\Delta^2} \ln\left( \frac{4K^2}{P_{e,k}} \right) \implies P_{e,k} \leq 4K^2 e^{-k} \tag{30}$$

which together with the requirement $\Delta < \frac{J_1 - J_2}{2N}$ of Lemma 1 completes the proof. $\qquad\square$

## 8.4 Proof Of Theorem 2

*Proof.* Let $z, z' \in Z$. Define for each $z$

$$\mathcal{N}_z = \{n \mid \bar{a}_n \neq a_n \text{ or } u_n = 0 \text{ or } M_n = D\}. \tag{31}$$

This is the set of players for which the transition of $M_n$ is governed by (11). Compared to [24], our dynamics have a different transition probability $P_{zz'}$ only when $z$ has a non-empty $\mathcal{N}_z$. For each $\mathcal{N}_z \subseteq \mathcal{N}$ we have

$$\lim_{\varepsilon \to 0^+} \frac{\prod_{n \in \mathcal{N}_z} \frac{u_n}{u_{n,\max}} \varepsilon^{u_{n,\max} - u_n}}{\varepsilon^{\sum_{n \in \mathcal{N}_z}(u_{n,\max} - u_n)}} = \prod_{n \in \mathcal{N}_z} \frac{u_n}{u_{n,\max}}. \tag{32}$$

Hence, in the limit $\varepsilon \to 0^+$, the ratio between the transition probabilities in our dynamics and those of [24] is either $\prod_{n \in \mathcal{N}_z} \frac{u_n}{u_{n,\max}} \leq 1$ or one. We conclude that each transition either has the same resistance as in [24] or it is impossible since $u_n = 0$ for some $n$. From any $z \in C_0$ there is a path with resistance $c$ to $D$, where a content player explores and becomes discontent. From any $z \in D$ to any $z \in C_0$ there is a path where all discontent players become content, which has resistance $\sum_n (u_{n,\max} - u_n(z))$ with $\{u_n(z)\}$ as the utilities in $z$. Therefore, the path from any $z$ to the maximizers of $\sum_{n \in \mathcal{N}} u_n$ (which are in $C_0$) has the same resistance as in [24] (it is the same path). Since all other paths have the same resistance as in [24] or do not exist, the maximizers of $\sum_{n \in \mathcal{N}} u_n$ remain the only stochastically stable states, as in [24]. From Lemma 4 we know that the maximizer of $\sum_{n \in \mathcal{N}} u_n$ is unique with probability 1. $\square$

## 8.5 Proof Of Lemma 5

*Proof.* Denote the optimal state by $z^* = \left[\boldsymbol{a}^{k*}, C^N\right]$. From Theorem 2 and the definition of a stochastically stable state, we know that for a small enough $\varepsilon$ we have $\pi(z^*) \geq \frac{2}{3}$. Denote the length of the part of the GoT phase where counting (of (6)) took place by $L \triangleq \lfloor c_2 (1 - \rho) k^{1+\delta} \rfloor$. In the end of the GoT phase, each player picks the action that she played most of the times she was content. If the strategy profile $\boldsymbol{a}^{k*}$ was played more than $\frac{L}{2}$ of the time, each player played the corresponding action at least half of the time. Hence, the probability that a strategy profile other than $\boldsymbol{a}^{k*}$ would be picked is lower than the probability that $\boldsymbol{a}^{k*}$ has been played less than $\frac{L}{2}$ of the time. We bound this probability using [27, Theorem 3]. Our function is $f(z) = I(z = z^*)$, that counts the number of visits to the optimal state. Note that the events $I(z(t) = z^*)$ are not independent but rather form a Markov chain. Hence, Markovian concentration inequalities are required.

We denote by $T_m\left(\frac{1}{8}\right)$ the mixing time of $Z$ with an accuracy of $\frac{1}{8}$. We define for the initial distribution $\varphi$ on $Z$ (after $d_g = \lceil \rho c_2 k^{1+\delta} \rceil$ turns in the $k$-th GoT Phase),

$$\|\varphi\|_\pi \triangleq \sqrt{\sum_{i=1}^{|Z|} \frac{\varphi_i^2}{\pi_i}}. \tag{33}$$

By choosing $\eta = 1 - \frac{1}{2\pi(z^*)}$ (so $0 < \eta < 1$ when $\pi(z^*) > \frac{1}{2}$), we obtain the following bound for a small enough $\varepsilon$

$$\Pr\left(\sum_{\tau=1}^L f(z(\tau)) \leq (1 - \eta)\pi(z^*)L\right) \leq \Pr\left(\sum_{\tau=1}^L I(z(\tau) = z^*) \leq \frac{c_2(1-\rho)k^{1+\delta}}{2}\right) \leq$$

$$c\|\varphi\|_\pi e^{-\frac{\left(1 - \frac{1}{2\pi(z^*)}\right)^2 \pi(z^*)c_2(1-\rho)}{72 T_m\left(\frac{1}{8}\right)} k^{1+\delta}} \underset{(a)}{\leq} c\|\varphi\|_\pi e^{-\frac{c_2(1-\rho)}{1728 T_m\left(\frac{1}{8}\right)} k^{1+\delta}} \tag{34}$$

where $c$ is some constant. Note that $\pi\left(1 - \frac{1}{2\pi}\right)^2 = \pi - 1 + \frac{1}{4\pi}$ is monotonically increasing for all $\frac{1}{2} < \pi < 1$. Since for a small enough $\varepsilon$ we have $\pi(z^*) \geq \frac{2}{3}$, (a) follows by substituting $\pi(z^*) = \frac{2}{3}$.

There is a tradeoff regarding $\|\varphi\|_\pi$. Starting from an arbitrary initial condition, $\|\varphi\|_\pi$ can be large. By dedicating the first $d_g$ turns of the GoT phase to letting $Z$ approach its stationary distribution, and starting to count the visits to $z^*$ only afterwards, we can reduce $\|\varphi\|_\pi$ significantly, at the cost of $d_g$ turns less for estimating $z^*$. Optimizing over $d_g$ (or $\rho$) and $\|\varphi\|_\pi$ can improve the constants of the bound (34). $\square$