[Reviews · NeurIPS 2018]

Reviewer 1



This paper proposed a distributed algorithm for a multi-player game of resource allocation. The algorithm repeats three phases: exploration phase, GoT phase, and exploitation phase. The paper is easy to read. I don't have time to check all the details. But the considered the game is interesting itself, and the algorithm is intuitive to some extent. However, the bound log^2 hides a lot of factors (as seen in Eq.(23)), which makes the reader less easier to understand the bound more deeply. For example, how is the bound dependent on the \Delta, or (J1-J2)/N defined in Lemma 1? As I can see, if (J1-J2)/N is very small (compared to 1/sqrt(T)), the regret bound can become totally vacuous because one can never ensure Eq.(12) hold with high probability. Then what is the regret in this case? Note that in the traditional N=1 case, we can always get the regret bound of min{logT/Delta, sqrt{T*logT}}. So I think to make the analysis more complete, the author should also try to get some bound when Eq. (12) cannot be totally ensured. I find the game setting itself to be interesting, and I'm eager to see more following works in this direction. Other comments: - Proposition 1: should use \Omega notation for a lower bound instead of big O

Reviewer 2



This paper studies a distributed multi-armed bandits setting, where every round each of N players must choose one of K arms. If a player picks the same arm as some other player, they receive payoff zero; otherwise, they receive payoff drawn from some distribution (specific to that player and that arm). This can be thought of as learning a distributed allocation or matching from players to arms. The goal is to design a communication-free learning algorithm that maximizes the total overall utility of all the players (or alternatively, minimizes their regret with respect to the best fixed allocation). The authors design an algorithm which receives total regret of O(log^2 T). Their algorithm can be thought of as a distributed "explore then exploit"-style algorithm (similar to e.g. epsilon-greedy). The algorithm begins with a phase where all players pick random arms to learn their individual distributions, and concludes with a phase where they all repeatedly pick what they have decided is the optimal allocation. The technically interesting part of the algorithm is the intermediate phase, where the arms coordinate amongst themselves (without communication, just by playing the game) to pick a good allocation where no arm is picked by more than one player (this they refer to as the "game of thrones"). They accomplish this by inducing a Markov process between the players of the game which converges to a state where all players are playing according to the optimal allocation. The authors also perform some simulations on synthetic data showing that this algorithm outperforms the case where all players are independently running UCB1. This is a nice result. That said, it seems like the interesting part of this work is not really the learning component, but rather the coordination component -- in fact, the problem/solution still seems interesting even if all players know their exact expected rewards for each arm ahead of time. From a learning perspective, I would have preferred to see a UCB-style algorithm that explores and exploits at the same time (while somehow simultaneously coming to a consensus on what allocation to play). Also, it seems important in the analysis that the distribution of rewards for an arm has no point mass at 0, since the way in which players estimate the expected reward for an arm is to just take the average over all non-zero pulls (zero-pulls are assumed to be collisions with other players). Is this necessary to the analysis? One can imagine natural settings where it is not possible to distinguish via the reward alone whether the reward is small because of the distribution or because other players are also playing that arm (either because the distribution has a point mass at 0 or because collisions don't decrease the reward all the way to 0). Does some variant of this algorithm work in this setting? (it is not obvious from the analysis in the paper). The paper was well-written and easy to understand. I have not checked all the proofs in the supplementary materials, but the proof sketches appear sound.

Reviewer 3



This paper considers the distributed multi-player bandits problem without assuming any communication between the players. It gives a distributed algorithm and proves that it attains poly-logarithmic regret. - The theoretical result holds only asymptotically and even there it is not clear that it is optimal. - On the practical side, the algorithm is complicated, introduces additional hyper-parameters without any intuition on how to set them. The lack of systematic experiments does not convince the reader about the usefulness of this approach. 1. Is it possible to extend this theory to the case when K < N. 2. The statement of the main theorem is not clear. Please quantify what you mean by small enough \epsilon and large enough c_2. 3. The algorithm is of the epsilon-greedy style where we do some initial exploration to obtain good estimates of the rewards, which is not optimal in the standard MAB case. Is the log^2(T) regret bound tight for this particular problem? 4. It seems that it is not possible to extend this theory to the case where the rewards are Bernoulli since it is difficult to disambiguate between collisions and zero rewards. Please clarify this. 5. For the GoT phase, please explain Lemma 3 better and make it self-contained. Also, please better connect the algorithm to the descriptions in the theorems. 6. Lemma 5 is a statement of large enough k. This is quite dissatisfying. Is there a way to bound this and say that after this phase, the regret scales as \log^2(T). 7. The algorithm introduces additional hyper-parameters c1, c2, c3. How do you set these in the experiments and what are guidelines for doing so? 8. Is it possible to prove something stronger assuming some minimal amount of communication? *** After Rebuttal *** I have gone through the other reviews and the author response. And my opinion remains unchanged.